# Application of Isochoric Impregnation: Effects on Microbial and Physicochemical Parameters and Shelf Life of Strawberries Stored Under Refrigeration

**DOI:** 10.3390/foods14030540

**Published:** 2025-02-06

**Authors:** Sumeyye Atci, Cristina Bilbao-Sainz, Valerie S. McGraw, Jiayuan Li, Gary Takeoka, Tara McHugh, Boris Rubinsky

**Affiliations:** 1U.S. Department of Agriculture, Western Regional Research Center, 800 Buchanan St., Albany, CA 94710, USA; cristina.bilbao@usda.gov (C.B.-S.); valerie.mcgraw@usda.gov (V.S.M.); jiayuan.li@berkeley.edu (J.L.); gary.takeoka@usda.gov (G.T.); tara.mchugh@usda.gov (T.M.); 2Department of Mechanical Engineering, University of California, 6141 Etcheverry Hall, Berkeley, CA 94720, USA; rubinsky@berkeley.edu

**Keywords:** isochoric cold storage, impregnation, strawberry, shelf life

## Abstract

This study evaluates the effectiveness of isochoric impregnation during isochoric cold storage in extending the shelf life of strawberries. Strawberries in two different impregnation solutions—(1) sucrose solution and (2) sucrose solution containing calcium chloride (CaCl_2_) and ascorbic acid (AA)—were first stored in an isochoric chamber at −2 °C/48 MPa for 1 week, followed by refrigeration at 4 °C for 3 weeks. For comparison, strawberries were also immersed in each solution for 1 week at 4 °C then refrigerated at 4 °C for 3 weeks. Additionally, a control group of fresh strawberries was stored at 4 °C for 4 weeks. The quality of the strawberries was assessed at 1-week intervals throughout the storage period. Isochoric cold storage effectively inhibited microbial growth and reduced the weight loss of the strawberries compared to refrigerated storage. Also, isochoric cold storage resulted in strawberries that retained their color attributes, whereas refrigeration led to a significant change in their color attributes. Isochoric impregnated strawberries in the sucrose solution containing CaCl_2_ and AA showed better mechanical properties and higher nutrient contents (anthocyanins and ascorbic acid) than those impregnated in the sucrose solution, impregnated in the solutions at 4 °C, and refrigerated with no impregnation (control). The results demonstrated that isochoric cold storage in a solution containing sucrose, CaCl_2_, and ascorbic acid effectively maintained the quality of the strawberries, as indicated by parameters such as firmness, color retention, and reduced microbial growth. This method extended the strawberries’ shelf life by up to 4 weeks.

## 1. Introduction

Strawberries (*Fragaria × ananassa Duch.*) are among the world’s favorite fruits, cherished for their distinct flavor, nutritional value, health benefits, low calorie content, vibrant color, and versatility. They can be consumed fresh, frozen, or processed. However, their soft texture makes them highly vulnerable to tissue damage, moisture loss, microbial contamination, and physiological deterioration during postharvest handling. Once harvested, they are quickly contaminated by microorganisms, leading to reduced firmness, discoloration, and a much shorter shelf life. This makes strawberries one of the most delicate and perishable horticultural products [1]. Fresh strawberries have a very short shelf life of just 1–2 days at room temperature, and low-temperature storage is the most commonly used method for extending their shelf life [2]. However, strawberries can only be stored for approximately 7 days at around 5 °C due to their delicate skin and soft flesh [3].

Globally, postharvest losses of strawberries can reach up to 50%, severely affecting the economic returns for growers and the availability of this nutritious fruit to consumers [4]. The berry industry has acknowledged the urgent need to extend the initial quality of strawberries beyond the current constraints [5]. There is a growing need for alternative, non-damaging, innovative, and environmentally friendly methods to preserve strawberries [2]. Consequently, extensive research has been conducted worldwide to develop advanced storage and distribution technologies for strawberries [6]. Some typical technologies include freezing [7], modified atmosphere packaging [8,9], gamma irradiation [10,11], and synthetic fungicides [12]. Although these methods are effective, they have certain limitations and environmental concerns. The use of synthetic fungicides raises significant safety and health concerns due to chemical residues on the fruit. Gamma irradiation, though successful in controlling microbes, faces challenges related to environmental impact and consumer acceptance. Additionally, freezing is energy-intensive, leading to questions about its sustainability and cost-effectiveness for long-term storage. Modified atmosphere packaging is less harmful to the environment but can be expensive and complicated to implement. Given these challenges, there is an increasing need for innovative and sustainable postharvest preservation techniques that reduce economic losses and environmental impact while ensuring the quality and microbial safety of strawberries [13].

Isochoric cold storage is one of the latest techniques developed to extend the shelf life of fresh fruits and vegetables. It uses subfreezing isochoric storage temperatures associated with pressures higher than atmospheric levels to decrease microbial loads [14,15,16]. In isochoric cold storage, food is preserved at subfreezing temperatures within a fixed-volume (isochoric) chamber filled with a liquid solution. The freezing process follows the liquidus line on the phase diagram of the solution, with concurrent changes in pressure and temperature. At thermodynamic equilibrium, a significant volume of unfrozen liquid remains at any given temperature in the chamber [17]. This allows food products to be preserved without harmful ice formation if they are kept in the unfrozen portion of the system [18]. Unlike traditional freezing, which requires energy-intensive cooling to lower temperatures below the freezing point, isochoric systems maintain food at a constant pressure, using minimal energy to achieve preservation. This results in lower energy consumption, reduced operational costs, and a more environmentally sustainable alternative for food preservation. Isochoric cold storage is more energy-efficient than conventional freezing due to less ice formation and reduced temperature fluctuations experienced by the food [15]. Also, recent studies have shown that the pressure generated during isochoric freezing can facilitate the infusion of liquid solutions into the pores of food products [19,20]. As a result, isochoric impregnation is an effective method for rapidly introducing external liquids, including those containing active compounds, into the porous structures of food products at subfreezing temperatures.

The objective of this study was to evaluate isochoric impregnation during isochoric cold storage at −2 °C/48 MPa as a postharvest treatment to extend the shelf life of strawberries during the subsequent refrigerated storage at 4 °C. Several nutritional and physicochemical quality attributes such as color, pH, titratable acidity, total soluble solids, weight change, fungal decay, texture, ascorbic acid, and total anthocyanin content were measured during the storage period.

## 2. Materials and Methods

### 2.1. Materials

Fresh organic strawberries (*Fragaria × ananassa Dutch*), sucrose, and food-grade calcium chloride (CaCl_2_) were purchased from a local supermarket. Only strawberries with similar size (approximately 30–35 mm in diameter) and maturity stage (commercially ripe, exhibiting a bright red color and firm texture) were selected. The strawberries were thoroughly washed with tap water to remove surface dirt and residues. Strawberries were stored for a maximum of one day at 4 °C before the subsequent sample processing. While all the strawberries were sourced and stored under identical conditions to ensure consistency at the start of the trials (T_0_), it is important to note that immediate postharvest processing was not feasible due to the supply chain involved in acquiring the fruit. Strawberries were carefully sorted aseptically to discard over-ripe, damaged, deformed, or poor-quality fruits, ensuring a uniform sample in terms of size and color. Each sterile polyethylene bag was filled with 80 ± 10 g of strawberries (5–6 pieces) in the following two different impregnation solutions: (a) sucrose solution containing 11% sucrose and (b) solution containing 1% CaCl_2_, 1% ascorbic acid, and 7.5% sucrose. CaCl_2_ was used to reduce softening, and the 1% concentration was selected as the maximum salt concentration that did not result in a salty flavor, based on sensory testing conducted with a panel of evaluators. In these preliminary tests, participants were asked to rate the intensity of the salty flavor in different concentrations of CaCl_2_, and the 1% concentration was determined to be the highest level that was still acceptable based on taste preferences. Ascorbic acid was used to prevent enzymatic browning, and the 1% concentration was chosen to ensure no alteration of the strawberry flavor, based on sensory testing where participants evaluated the flavor of the strawberries treated with varying concentrations of ascorbic acid. The simultaneous application of calcium salts and ascorbic acid improved fruit color and enhanced vibrancy and stability [21]. The concentrations of 7.5% and 11% sucrose were used to achieve osmotic balance between the strawberries and the impregnation solutions [22]. Concentrations are reported as mass percentages (*w*/*v*), expressed as ‘%’.

### 2.2. Postharvest Isochoric Cold Storage

Isochoric chambers (BioChoric Inc., Bozeman, MT, USA), constructed from aluminum-7075 with a type-II anodized coating, with a total volume capacity of 1.5 L and pressure-rated up to 210 MPa, was used for the isochoric cold storage (ICS). Each chamber was equipped with a pressure gauge to monitor the pressure over time. Initially, a steel nut, serving as an ice nucleating agent, was placed at the base of the isochoric chamber to ensure that ice formation occurred away from the sample bags. Subsequently, a plastic spacer apparatus was inserted into the chamber, and the samples were transferred onto the plastic apparatus, with three bags placed in each chamber. The chambers were then sealed after being filled with water. The isochoric chambers were subsequently stored in chest freezers (Magic Chef 123 Model #HMCF9W3, MC Appliance Corporation, Wood Dale, IL, USA) at −2 °C for 7 days. The freezing temperature of −2 °C was chosen based on preliminary work to avoid high pressures that would result in cell damage [23]. In an isochoric environment, the temperature and the volume of unfrozen liquid in the chamber are correlated. At the triple point (−22 °C, 210 MPa for pure water), about 45% of the initial volume remains unfrozen. As the temperature rises, this percentage increases, reaching 90% of the initial volume at −2 °C, 20 MPa [24]. By carefully controlling temperature and pressure, strawberries can be kept within the unfrozen region of the system, ensuring they remain preserved without ice formation [18]. Following the ICS, the chambers were relocated to a fridge at 5 °C and left overnight to allow the ice within the chamber to melt.

### 2.3. Experimental Protocol

The following five different methods were used to preserve fresh strawberries for up to 4 weeks: refrigeration without impregnation solution at 4 °C for 4 weeks (RF); refrigeration in sucrose solution at 4 °C for one week and refrigeration without solution at 4 °C for 3 additional weeks (RF + S); refrigeration in sucrose + CaCl_2_ + ascorbic acid (AA) solution at 4 °C for one week and refrigeration without solution at 4 °C for 3 additional weeks (RF + C); isochoric cold storage in sucrose solution at −2 °C for one week and refrigeration without solution at 4 °C for 3 additional weeks (ICS + S), and isochoric cold storage in sucrose + CaCl_2_ + AA solution at −2 °C for one week and refrigeration without solution at 4 °C for 3 additional weeks (ICS + C).

For the refrigerated treatment without a solution, three clamshell containers, each containing 80 ± 10 g of strawberries, were stored at 4 °C and 95% relative humidity for 4 weeks (Figure 1). For the refrigerated treatments with impregnation solutions, 80 ± 10 g of strawberries placed in pouches containing 120 mL of the solutions were stored at 4 °C for 7 days. After 7 days, the solution was removed, and the strawberries were kept refrigerated in clamshell containers at 4 °C for 3 additional weeks. For ICS, 3 pouches containing 80 ± 10 g of strawberries in 120 mL of each solution were stored under isochoric conditions at −2 °C/48 MPa for 7 days. After 7 days, strawberries were taken out of pouches and refrigerated in clamshell containers at 4 °C for 3 additional weeks. Physical, biochemical, and microbial properties were examined at one-week intervals. Some of the fruits were tested for microbiology, color, and firmness. Strawberry juice squeezed from 2 to 3 strawberries was pooled from each pouch and used to measure pH, titratable acidity, and total soluble solids. The rest of the fruits were stored at −80 °C for the determination of the ascorbic acid and anthocyanin contents.

### 2.4. Enumeration of Microbial Populations

Microbiological evaluations of the strawberries were carried out on day 0 (fresh) and after weeks 1, 2, 3 and 4. Two to three fruits from each replicate were randomly chosen and cut into pieces. Strawberry pieces (10 g) were placed in a sterile bag (Whirl-Pak, Nasco, Fort Atkinson, WI, USA) and then homogenized in 10 mL of PBS (pH 7.4) using a stomacher (Seward Stomacher 80 Biomaster Laboratory Blender, Seward Inc., Bohemia, NY, USA) at a “normal” speed (equivalent to 230 rpm) for 60 s. A 10-fold serial dilution was made from the homogenate, and 1 mL of diluted homogenate from each dilution was inoculated onto plate count agar (PCA; Sigma Aldrich, St Louis, MO, USA) for total plate count (TPC). The plates were incubated at 30 °C for 48 h, and the colonies (about 25–250) were counted. To count the yeast and mold (Y&M), 1 mL of the homogenate was plated similarly onto potato dextrose agar (PDA; Difco, Detroit, MI, USA) and incubated at 25 °C for 72 h. Microbial counts were converted to log CFU per gram of sample for both the TPC and Y&M. Each sample was analyzed in triplicate.

### 2.5. Measurement of Physicochemical Parameters

#### 2.5.1. Fungal Decay

Fungal decay was monitored through daily visual inspections during the storage period, regardless of infection severity [25]. Strawberries displaying surface mycelial growth were classified as decayed, and the results were reported as the percentage of infected fruits.

#### 2.5.2. Weight Change

Weight change (%) was determined for all strawberries in the pouch using the following formula:$$Weight\;change=\frac{m_{0}-m_{1}}{m_{0}}\times 100$$
where *m*_1_ is the weight of the fruit on each sampling day, and *m*_0_ is the original weight of the fresh fruit.

#### 2.5.3. Moisture Content, pH, Titratable Acidity (TA), and Total Soluble Solids (TSSs)

The moisture content was determined by placing each sample in an oven (Fisher Scientific Isotemp model 750 F, Pittsburgh, PA, USA) at 72 °C for 24 h and measuring the resulting change in mass (AOAC, 1990). The pH, titratable acidity (TA), and total soluble solids (TSSs) were measured using juice extracted from a mixture of 2–3 strawberries from each pouch. The pH was obtained using a pH meter (Hanna instruments, Woonsocket, RI, USA). Total titratable acidity (TA) (expressed as citric acid %) was determined potentiometrically by adding 0.01 N NaOH to the titration end point (pH of 8.2) of a 2 mL sample. TSS (expressed as %) was determined by measuring the refractive index of the juice using a digital refractometer (Maselli LR-01, Masseli Misure s.p.a., Parma, Italy). All experiments were performed in triplicate.

#### 2.5.4. Surface Color

Two or three randomly selected fruits from each pouch were used for color measurement using a digital colorimeter (CM-3500d, Konica-Minolta Inc., Ramsey, NJ, USA) equipped with a 3 mm diameter CM-A195 target mask. Color changes were quantified in the *L**, *a**, *b** color space. *L** refers to the lightness and ranges from black  =  0 to white  =  100. A negative value of *a** indicates green, while a higher positive value indicates red. A negative value of *b** indicates blue, whereas a higher positive value indicates yellow. Before measuring the surface color of the strawberries, the meter was calibrated using a standard white and black plate supplied by the manufacturer.

Chroma (*C**) in Equation (1) represents the intensity or saturation of the color, while the hue angle (*h*, in °) in Equation (2) defines the qualitative attribute of color shades, where 0° corresponds to reddish-purple and 180° to bluish-green [26].(1)$$C*=\sqrt{{a*}^{2}+{b*}^{2}}$$(2)$$h=arctan\frac{b*}{a*}\times \frac{180}{\pi }\;(\mathrm{degrees})+180{}^{\circ}\;\mathrm{when}\;a*<0\;\mathrm{and}\;b*>0\;\mathrm{or}\;b*<0$$

#### 2.5.5. Mechanical Properties

Two or three strawberries from each pouch were randomly selected and used for firmness measurements using a texture analyzer (Stable Microsystems Ltd., TA-XT2i, Surrey, UK) equipped with a 3 mm diameter stainless steel cylinder probe. Pre-test, test, and post-test speeds were set at 5 mm/s, 1 mm/s, and 5 mm/s, respectively. Firmness was determined by measuring the peak compression force exerted during the penetration of the strawberry. Each strawberry fruit was placed horizontally on the stationary platform of the analyzer, and the probe pressed on the fruit surface at the equatorial zone.

### 2.6. Total Anthocyanins Content

Total anthocyanins were estimated using a spectrophotometric pH-differential method following the procedure of Lee et al. [27]. A pH 1.0 buffer was prepared by adjusting 0.025 M KCl (Fisher Scientific) with 1 M HCl (Fisher Scientific). Similarly, a pH 4.5 buffer was prepared by adjusting a 0.4 M CH_3_CO_2_Na·3H_2_O (Sigma-Aldrich, St. Louis, MO, USA) solution with HCl. Strawberries, previously frozen at −80 °C, were flash-frozen in liquid nitrogen for 1 min and then crushed into fine pieces using a hammer. One gram of fruit tissue was added to a centrifuge tube with 20 mL of the pH 1.0 or 4.5 buffer. The tubes were incubated overnight at 5 °C in darkness for 18–24 h and then centrifuged (15,600 rpm) at 4 °C for 15 min. Absorbance of these two solutions was measured at both 510 and 700 nm, using a UV–VIS spectrophotometer (Shimadzu UV 1280, Shimadzu Scientific Instruments, Inc., Columbia, MD, USA). Concentration of anthocyanins was calculated by using Equation (3) as described in the following equation [27]:(3)$$Anthocyanin\;pigment\;\left(cyanidin-3-glucoside\;equivalents,\frac{\mathrm{mg}}{L}\right)=\frac{A\times MW\times DF\times {\;10}^{3}}{\varepsilon \times 1}\;$$
where *A* is absorbance, molar absorptivity (ε) is 26,900 (L·cm^−1^·mol^−1^), *MW* is the molecular weight of cyanidin-3-glucoside (449.2), and *DF* is the dilution factor in the experiment. Absorbance (*A*) was calculated using the measured absorbances at 510 and 700 nm for pH 1.0 and pH 4.5 solutions as in Equation (4):(4)$$A={\left(A_{510}-A_{700}\right)}_{pH\;1.0}-{\left(A_{510}-A_{700}\right)}_{pH\;4.5}$$

Anthocyanin content (mg/L) was then converted to mg/100 g fresh sample (dry matter).

### 2.7. Ascorbic Acid Determination

The ascorbic acid (AA) content was determined using HPLC, following a modified version of the method by Bilbao-Sainz, Sinrod, Powell-Palm, Dao, Takeoka, Williams, Wood, Ukpai, Aruda, Bridges, Wu, Rubinsky, and McHugh [23]. Three grams of fruit tissue was mixed with 7.5 mL of metaphosphoric acid solution (30 g metaphosphoric acid, 0.5 g EDTA, and 80 mL glacial acetic acid, diluted to 1 L with deionized water) then centrifuged (10,000 rpm) at 4 °C for 15 min. A solid-phase extraction cartridge (Bond Elut C18, 500 mg, 3 mL, Agilent Technologies, Wilmington, DE, USA) was preconditioned with 2 mL of acetonitrile and 3 mL of distilled water before being used to filter the supernatant. The AA content was then analyzed by injecting 50 μL of the filtered sample into an Agilent 1100 series HPLC system equipped with a diode array detector set to 265 nm. Separation was performed on an ICSep ICE-ION-300 (300 × 7.8 mm) column with a corresponding guard column (Transgenomic, Inc., San Jose, CA, USA). The mobile phase consisted of 20 mM H_2_SO_4_ at a flow rate of 0.3 mL/min. The AA content was quantified using a standard calibration curve prepared with the AA standards.

### 2.8. Statistical Analysis

Each experiment was performed in triplicate, and the results are expressed as the mean ± standard deviation. Microsoft Excel was used for statistical analysis. Significant differences between samples were assessed by analysis of variance (ANOVA) and Tukey’s post hoc test means comparison test at a 95% confidence level.

## 3. Results and Discussion

### 3.1. Microbiological Evaluation

Table 1 shows the microbial load of the strawberries during storage. The initial total plate count (TPC) and yeast and mold (Y&M) were both 3.9 ± 0.3 log CFU/g. After 4 weeks of refrigeration, the RF, RF + S, and RF + C samples had significantly increased TPC values of 4.8 ± 1.2 log CFU/g (*p* < 0.05), 7.3 ± 0.3 log CFU/g (*p* < 0.05), and 6.3 ± 0.5 log CFU/g (*p* < 0.05), respectively, and increased Y&M counts of 4.8 ± 0.3 log CFU/g (*p* > 0.05), 7.2 ± 0.3 log CFU/g (*p* < 0.05), and 7.0 ± 0.5 log CFU/g (*p* < 0.05), respectively. A previous study found that after 18 days of refrigerated storage of the strawberries at 4 °C, the TPC increased from 2.9 to 3.6 CFU/g and the Y&M counts increased from 3.6 to 4.3 CFU/g [28]. From Table 1, the RF + S strawberries had higher TPC than the RF + C strawberries at weeks 1 and 4. Lower microbial populations in RF + C strawberries could be attributed to the antimicrobial properties of ascorbic acid (AA), which has been used to reduce microbial populations in fruits and vegetables [28]. The use of AA, along with certain organic acids like lactic acid, could serve as a promising preservative to prevent the growth of *E. coli* O157 in Brain Heart Infusion (BHI) broth and carrot juice [29]. Also, a 1.5% solution of AA applied as a 2 min immersion for jackfruit was found to be as effective as chlorine in significantly reducing aerobic bacteria, coliforms, and yeasts and molds after 3 h of treatment [30].

The addition of sucrose and calcium chloride (CaCl_2_) can influence microbial populations on strawberries. Sucrose, as a fermentable sugar, may promote the growth of certain microorganisms, including yeasts and molds, which utilize sugars as a primary energy source [31]. The reduction of microorganisms in strawberries treated with CaCl_2_ may be attributed to the ability of calcium salts to lower intracellular pH or reduce water activity. These effects create a protective antimicrobial barrier, limiting the growth of foodborne pathogens in the fruit [32].

The microbial dynamics of strawberries are influenced by various factors, including microbial–microbial interactions and environmental conditions such as humidity. High humidity facilitates the growth of molds, while a reduction in moisture content over time may suppress microbial proliferation, potentially explaining the lack of an increase in microbial counts by week 4 [33]. Additionally, the refrigerated strawberries stored for 4 weeks showed a similar decay rate but a lower CFU count compared to those stored for 3 weeks. This decrease in viable fungal spores during extended storage may result from environmental stress or microbial competition. However, advanced ripening and fruit softening over time may increase susceptibility to fungal invasion and microbial growth [34].

The quality of strawberries during storage is often compromised due to their highly perishable nature, making them susceptible to postharvest diseases caused by bacteria, yeasts, and fungal infections. In the current study, isochoric cold storage (ICS) kept microbial populations below the detection limit (1 CFU/g) in strawberries within both solutions, except for Y&M at week 4 (Table 1). Our findings are consistent with the previous studies on ICS-treated blueberries [35], raw milk [36], pomegranate fruit [37], carrot juice [38], and orange juice [39]. These studies demonstrated a reduction in mesophilic aerobic bacteria and Y&M populations during storage.

### 3.2. Visible Fungal Growth

The RF sample began showing visible signs of fungal decay after 1 week of storage at 4 °C (Figure 2). After 2 weeks of storage, 50% of RF strawberries, 50% of RF + C, and 87% of RF + S showed mold infection. Chen et al. [40] indicated that a 1% CaCl_2_ treatment slightly reduced the decay rate of strawberry fruits during the initial 10 days of storage. This effect was attributed to calcium’s ability to enhance fruit tissue integrity, providing protection against fungal contamination [41]. By week 4, the RF, RF + S, and RF + C strawberries showed 100% infection. In contrast, the ICS + S and ICS + C strawberries exhibited no visible signs of fungal decay throughout the 4-week storage period.

Visible signs of decay mark the point when strawberries are no longer acceptable for consumption [42]. While fungicide applications during the growing cycle are the traditional method for controlling postharvest decay, their use is increasingly questioned due to sustainability and safety concerns, with bans in many countries [43]. Alternatives like pulsed light [25], hypobaric treatment [44], or ultrasound [45] help to slow decay but do not fully prevent spoilage. In contrast, the findings of this study demonstrate that the pressure applied during ICS successfully inactivated fungal growth on strawberries after a week. Furthermore, fungal growth remained undetectable during the subsequent 3 weeks of refrigerated storage at 4 °C.

The classification of fruit decay as simply “rotten” and “non-rotten” is a limitation in our study. A more detailed decay assessment method will provide a more comprehensive understanding of the decay process and enable a better evaluation of treatment effectiveness.

### 3.3. Weight Changes

The changes in strawberry weight over the storage period are depicted in Figure 3. After 1 week of storage at 4 °C, RF and RF + C strawberries lost weight, while RF + S strawberries showed weight gain. As shown in Figure 3, the ICS strawberries in both impregnation solutions experienced weight gain after 1 week due to the solution penetrating the porous tissue and intercellular spaces through a hydrodynamic mechanism consisting of capillary action and pressure gradients [46]. Throughout storage, the refrigerated strawberries (RF, RF + S, and RF + C) experienced significant weight loss, reaching up to 80% by week 4. This was primarily due to moisture loss from evaporation between the fruit tissue and the surrounding environment as well as the respiration processes [1]. Given their extremely thin skin, strawberry fruits are highly prone to rapid water loss [47,48]. However, strawberries subjected to isochoric impregnation had limited weight loss compared to the refrigerated samples. This can be attributed to the mechanisms of isochoric impregnation, which involve pressure-induced mass transfer, allowing the liquid solution to penetrate the fruit’s porous structure [20]. This process not only enhances the retention of moisture within the fruit but also creates a barrier to water loss, effectively mitigating dehydration.

### 3.4. Effects of Preservation Method on Moisture Content, pH, Titratable Acidity (TA), and Total Soluble Solids (TSSs)

The moisture content, pH, titratable acidity (TA), and total soluble solid (TSS) parameters of the strawberries are presented in Table 2. The moisture content of all the strawberries in solution either slightly increased in value or remained stable after 1 week, which could be due to an infusion of the solutions. However, the RF + S and RF + C samples had significantly lower moisture contents by weeks 3 and 4, albeit not as low as those of the RF sample. ICS impregnated strawberries exhibited less moisture loss than refrigerated strawberries (RF + S and RF + C), regardless of the solution used. The pH values of the RF + C and ICS + C samples were significantly lower than the others due to acidic nature of ascorbic acid [49]. All samples showed a decrease in pH value throughout storage. The titratable acidity (TA) of fresh strawberries was 1.8%. All the strawberry samples showed an increase in TA during storage, in agreement with Vicente, Martínez, Civello, and Chaves [34]. In addition, the TSS of all the strawberry samples significantly increased in value during storage. This was attributed to moisture loss, the breakdown of complex sugars into simpler sugars, the degradation of cell walls, and the overall decay of the fruit [50]. After 4 weeks, the RF + S and RF + C strawberries became too small and shriveled, making it impossible to perform pH, TA, and TSS analyses.

### 3.5. Color and Appearance

The color parameter values (lightness *L**, redness *a**, yellowness *b**, chroma *C**, hue angle *h°*) of the strawberries are presented in Table 3. All the strawberries showed no significant changes in *L** values, except for the RF + S and RF + C samples after 4 weeks, which showed increases in lightness values. This was probably due to the presence of mold. Strawberries that lost more moisture (RF + S and RF + C) during storage showed increased darkening, primarily due to oxidative browning reactions [51,52].

The redness (*a**) values of RF strawberries remained relatively stable during the first 3 weeks of storage but significantly declined in value after 4 weeks (*p* < 0.05). Also, the chroma (*C**) decreased and the hue angle (*h**) increased in value. Storing strawberries in solutions for 1 week did not significantly change the redness values (*p* > 0.05). However, the samples stored in sucrose solution after 1 week had significantly lower yellowness (*b**) values than those of the fresh strawberries, although their *b** values increased in value during storage and eventually became similar to those of the fresh strawberries (*p* > 0.05). The RF + C sample also had higher *a** values compared to the RF + S sample. The RF + C and RF + S samples had significantly lower redness values by week 4 (*p* < 0.05), leading to decreases in *C** and increases in *h** values. This may be attributed to the presence of mold and reduced levels of anthocyanin pigments responsible for the red color in strawberries [53,54].

The ICS + S and ICS + C samples showed no significant changes in the redness (*a**) values during 4 weeks of storage (*p* > 0.05). In contrast to the refrigerated samples, the ICS strawberries showed higher *a** and *C** values at week 4 than at shorter storage times. The ICS + C sample had higher *a** values than those of the ICS + S sample, resulting in higher *C** values. These results suggest possible interactions between ascorbic acid (AA) and the anthocyanins. Gliemmo et al. [55] previously demonstrated that AA effectively preserved the red color of pumpkin. Also, the color changes in apples impregnated with solutions containing AA were less pronounced compared to those treated with sucrose solutions [56]. Calcium in the solution may also influence color values. Studies have shown that calcium plays a role in preserving color stability by strengthening cell walls and reducing enzymatic activity. For instance, Bilbao-Sainz, Millé, Chiou, Takeoka, Rubinsky, and McHugh [35] and McGraw et al. [57] reported that the isochoric impregnation of blueberries with blueberry juice and calcium solution exhibited significantly higher *a** values compared to fresh and refrigerated samples, showing a more vibrant blue color as well.

The appearance of the strawberries during storage is illustrated in Figure 4. After one week, mold growth occurred on some RF strawberries, while those refrigerated in both solutions (RF + S and RF + C) retained their color and exhibited no signs of mold on their surface. However, by the second week, mold growth occurred in the RF + S and RF + C strawberries as well. By week 4, the RF, RF + S, and RF+ C strawberries were completely covered with mold. In contrast, the isochoric-cold-stored strawberries (ICS +S and ICS + C) appeared darker by week 4 but showed no signs of mold growth.

### 3.6. Texture

The hardness values of the strawberries are presented in Table 4. The RF strawberries showed a reduction in hardness during the refrigerated storage, especially between week 1 and week 2. This decrease in firmness is mainly related to biochemical alterations at the cell wall, middle lamella, and membrane levels due to the activity of pectin methylesterase, an enzyme that hydrolyzes pectin, leading to structural breakdown [58]. However, the firmness of the RF sample during storage increased to a higher level than that of the fresh sample, which had been attributed to an increase in pectin viscosity [41]. Additionally, water loss from the outer layer can increase fruit density, lower gas permeability, reduce oxygen levels, and elevate internal carbon dioxide concentrations, which may contribute to the firmer texture [59].

The refrigerated strawberries (RF + S and RF + C) also showed an increase in hardness throughout the storage period, likely attributed to a significant loss of internal water content in the cells [60]. However, the samples had high standard deviations since certain strawberries remained quite firm while others were very soft. The breakdown in texture may have been caused by the increase in aerobic microbial counts [61], which could have led to the production of pectinolytic enzymes responsible for tissue softening [62].

The ICS strawberries were harder than the RF strawberries for the same solution. Also, the strawberries impregnated with the solution containing sucrose, CaCl_2_, and AA were harder than the strawberries impregnated only with the sucrose solution. The increase in hardness was attributed to the addition of exogenous calcium ions (Ca^2+^) to the strawberry fruit [63] via impregnation with pressure. Calcium ions (Ca^2+^) are essential for maintaining fruit quality by inhibiting the activity of polygalacturonase (PG), an enzyme responsible for breaking down cell wall components such as pectins [64]. Furthermore, Ca^2+^ binds with demethylesterified pectin backbones to form a pectin–Ca^2+^ network, which strengthens the mechanical properties of the cell wall and helps preserve fruit texture [65,66,67]. Koushesh Saba and Sogvar [68] reported that Ca^2+^ helped maintain and enhance the integrity and mechanical properties of the cell wall, effectively preventing the softening of fruits.

### 3.7. Total Anthocyanin Content

Figure 5 shows the total anthocyanin content of the strawberries during storage. The total anthocyanin content of the fresh strawberries was 20.2 ± 4.4 mg/g dry matter (d.m.). The strawberries stored in sucrose solution for one week (RF + S and ICS + S) did not show a significant change in the total anthocyanin content (*p* > 0.05), whereas the strawberries in the sucrose + CaCl_2_ + AA solution (RF + C and ICS + C) showed a significant decrease in the total anthocyanin content (*p* < 0.05). The high anthocyanin content in the RF + S samples can be attributed to the sucrose treatment, which stimulates anthocyanin accumulation by upregulating the expression of genes involved in anthocyanin biosynthesis [69]. The CaCl_2_ treatment positively influenced the retention of monomeric anthocyanins during storage by facilitating pectin–anthocyanin binding [70]. The presence of ascorbic acid accelerated anthocyanin degradation and led to a loss of color, indicating a direct interaction between the two molecules [71]. The lower pH in the strawberries impregnated with the AA solution could also contribute to the reduced anthocyanin content, since the stability of anthocyanins is influenced by pH [72].

The total anthocyanin content of the refrigerated samples significantly decreased during storage (*p* < 0.05). The anthocyanin degradation might be associated with water loss during storage, leading to physiological stress and accelerating fruit senescence [52]. Water loss led to membrane disintegration and leakage of cellular contents [73], both of which contributed to the decrease in anthocyanin concentration. In addition, the increase in enzyme activity, such as polyphenol oxidase, may also have contributed to the reduction in anthocyanin content in the strawberries during storage [74].

The ICS samples showed significantly lower anthocyanin contents after one week compared to the refrigerated samples. This initial decrease can likely be attributed to the effects of pressure and impregnation during isochoric treatment, which may introduce physical stress and promote anthocyanin degradation [37]. At week 4, the ICS samples exhibited significantly higher total anthocyanin contents compared to the RF, RF + S, and RF + C samples (*p* < 0.05), becoming redder and darker over time due to the synthesis of anthocyanins, the pigments responsible for the red color in strawberries [32].

### 3.8. Ascorbic Acid Content

The ascorbic acid (AA) content in the fresh strawberries was 442 ± 13 mg/100 g dry matter (d.m.), which was comparable to the values of 340–680 mg/100 g d.m. [75] and 490 mg/100 g d.m. [76] found in the previous studies. The RF, RF + S, and ICS + S strawberries had similar AA contents after one week (*p* > 0.05). However, the RF + C and ICS + C strawberries showed significant increases in AA contents after one week due to the presence of AA in the impregnation solutions (Figure 6).

CaCl_2_ treatment may help preserve AA in strawberries during storage by preventing the breakdown of antioxidants, such as AA [32,77]. The application of CaCl_2_ may have facilitated the formation of pectin–calcium complexes, which minimized cellular damage, maintained cell integrity, and reduced the leakage of intracellular components [35]. The rise in AA content during storage could have resulted from the ongoing biosynthesis of vitamin C, likely as a physiological reaction of the fruit cells to the stress caused by CaCl_2_ treatment [70]. Additionally, CaCl_2_ applications effectively slowed down the oxidation of AA by enhancing the activity of key catalytic enzymes involved in fruit biosynthesis thus helping to preserve AA levels [78,79,80,81].

All samples showed decreasing AA content during storage (Figure 6). The positive effects of CaCl_2_ impregnation were diminished when samples were stored in the refrigerator, likely due to a rise in respiration rate at higher temperatures. Increased fruit respiration can elevate CO_2_ levels, which has been shown to cause ascorbic acid loss [35]. However, the ICS + C sample had significantly smaller decreases in AA contents during weeks 3 and 4 (*p* < 0.05) compared to the RF + C sample.

## 4. Conclusions

This study examined the effects of isochoric cold storage (ICS) at −2 °C/48 MPa in combination with isochoric impregnation with sucrose solution or sucrose solution containing calcium chloride (CaCl_2_) and ascorbic acid on the quality of strawberries. The refrigerated strawberries at 4 °C experienced growth of mesophilic aerobic bacteria, yeasts, and molds over the 4-week storage period, whereas isochoric impregnation effectively inhibited the growth of these microorganisms over the same period. After 4 weeks, refrigeration at 4 °C resulted in significant weight loss in the strawberries, with reductions of 79% in the CaCl_2_ solution and 82% in the sucrose solution. In contrast, ICS helped minimize weight loss, with reductions of 68% in the CaCl_2_ solution and 60% in the sucrose solution during refrigerated storage. Also, ICS strawberries in the presence of CaCl_2_ and ascorbic acid showed better mechanical properties, color stability, and higher nutrient content (anthocyanins and ascorbic acid) than those in the sucrose solution or under refrigeration. Overall, ICS with sucrose, CaCl_2_, and ascorbic acid impregnation proved to be a highly promising postharvest technology for extending the shelf life of strawberries for up to 4 weeks. This study highlights the potential of ICS not only for improving the storage stability of strawberries but also as a sustainable alternative to conventional methods. Future research should focus on scaling up this technology and evaluating its feasibility for commercial applications, offering a pathway to reduce postharvest losses and meet the growing demand for longer-lasting, high-quality fresh produce.

## Figures and Tables

**Figure 1 foods-14-00540-f001:**
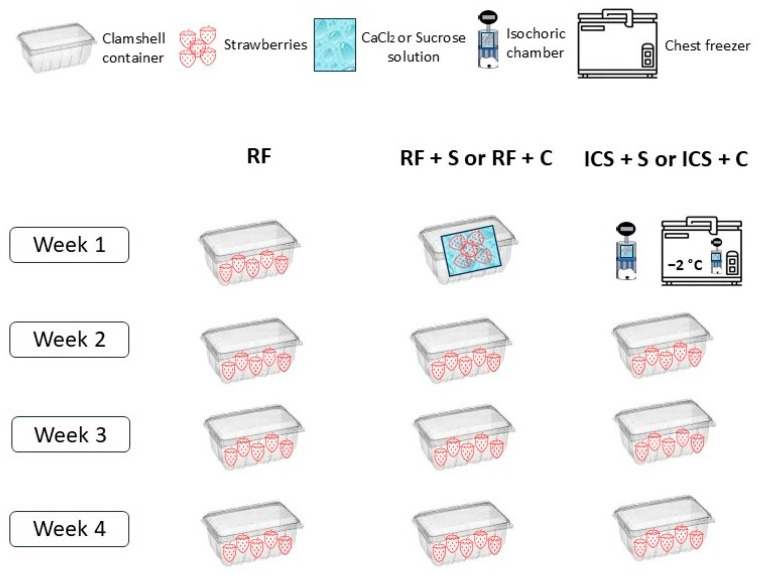
Experimental design.

**Figure 2 foods-14-00540-f002:**
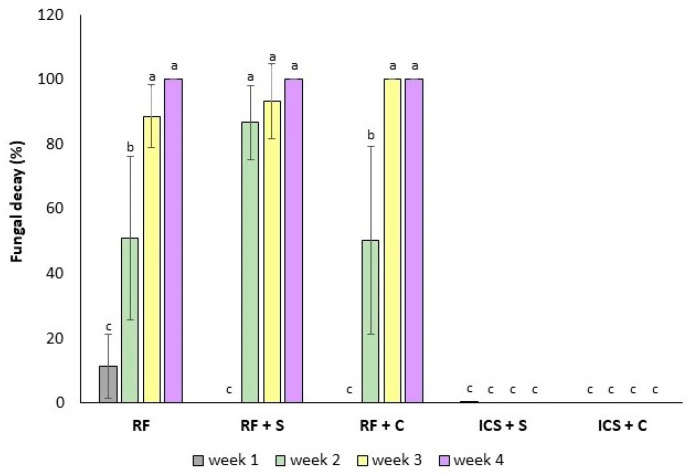
Fungal decay (%) of preserved strawberry. RF: refrigeration, RF + S: refrigeration with sucrose solution, RF + C: refrigeration with sucrose + CaCl_2_ + ascorbic acid (AA) solution, ICS + S: isochoric cold storage with sucrose solution, ICS + C: isochoric cold storage with sucrose + CaCl_2_ + AA solution. Values followed by the same letter (a–c) are not statistically different at *p* < 0.05.

**Figure 3 foods-14-00540-f003:**
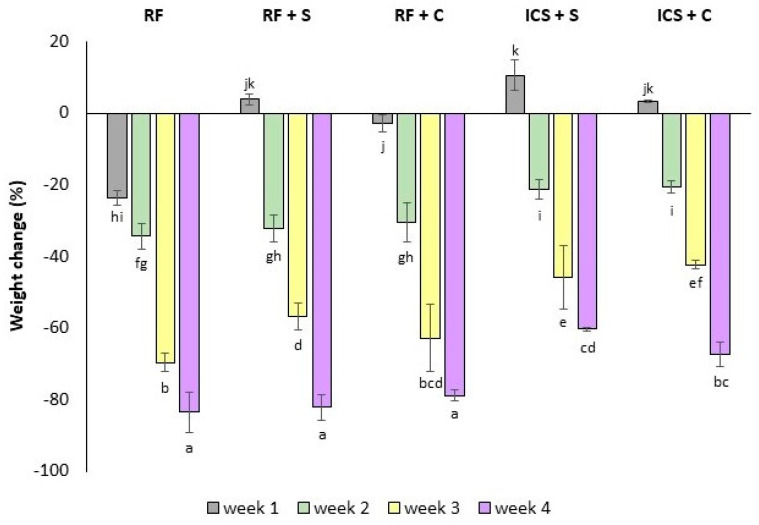
Weight change (g) of preserved strawberries. RF: refrigeration, RF + S: refrigeration with sucrose solution, RF + C: refrigeration with CaCl_2_ + ascorbic acid (AA) solution, ICS + S: isochoric cold storage with sucrose solution, ICS + C: isochoric cold storage with sucrose + CaCl_2_ + AA solution. Values followed by the same letter (a–k) are not statistically different at *p* < 0.05.

**Figure 4 foods-14-00540-f004:**
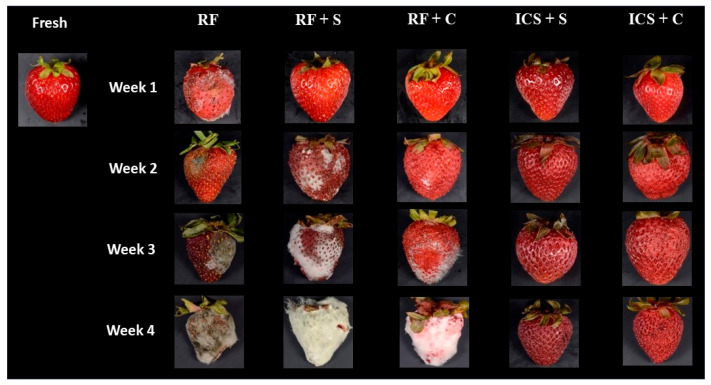
Appearance of fresh and preserved strawberries. RF: refrigeration, RF + S: refrigeration with sucrose solution, RF + C: refrigeration with CaCl_2_ + ascorbic acid (AA) solution, ICS + S: isochoric cold storage with sucrose solution, ICS + C: isochoric cold storage with sucrose + CaCl_2_ + AA solution.

**Figure 5 foods-14-00540-f005:**
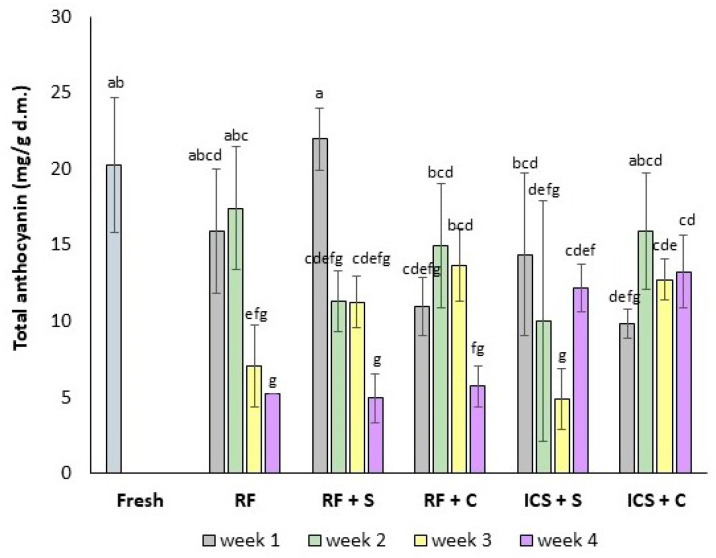
Total anthocyanin content (mg/g d.m.) in fresh and preserved strawberries. RF: refrigeration, RF + S: refrigeration with sucrose solution, RF + C: refrigeration with CaCl_2_ + ascorbic acid (AA) solution, ICS + S: isochoric cold storage with sucrose solution, ICS + C: isochoric cold storage with sucrose + CaCl_2_ + AA solution. Values followed by the same letter (a–g) are not statistically different at *p* < 0.05.

**Figure 6 foods-14-00540-f006:**
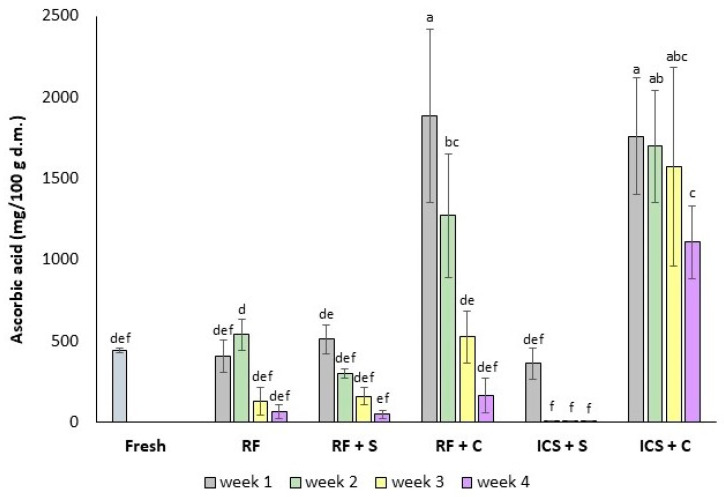
Ascorbic acid content (mg/100 g d.m.) in fresh and preserved strawberries. RF: refrigeration, RF + S: refrigeration with sucrose solution, RF + C: refrigeration with sucrose + CaCl_2_ + ascorbic acid (AA) solution, ICS + S: isochoric cold storage with sucrose solution, ICS + C: isochoric cold storage with sucrose + CaCl_2_ + AA solution. Values followed by the same letter (a–f) are not statistically different at *p* < 0.05.

**Table 1 foods-14-00540-t001:** Total plate count (TPC), yeast and mold (Y&M) in fresh and preserved strawberry. RF: refrigeration, RF + S: refrigeration with sucrose solution, RF + C: refrigeration with sucrose + CaCl_2_ + ascorbic acid (AA) solution, ICS + S: isochoric cold storage with sucrose solution, ICS + C: isochoric cold storage with sucrose + CaCl_2_ + AA solution.

Microbial Population	Fresh	Shelf Life (Weeks)	RF	RF + S	RF + C	ICS + S	ICS + C
Total plate count		1	4.2 ± 0.3 ^efg^	5.4 ± 0.7 ^cde^	4.2 ± 0.7 ^efg^	<1 ^h^ *	<1 ^h^
(TPC)	3.9 ± 0.3 ^g^	2	4.0 ± 1.1 ^fg^	6.6 ± 0.4 ^abc^	6.6 ± 1.4 ^abc^	<1 ^h^	<1 ^h^
(log CFU/g)		3	5.2 ± 0.2 ^def^	7.6 ± 0.3 ^a^	7.3 ± 0.5 ^ab^	<1 ^h^	<1 ^h^
		4	4.8 ± 1.2 ^ef^	7.3 ± 0.3 ^ab^	6.3 ± 0.5 ^bcd^	<1 ^h^	<1 ^h^
Yeast and mold		1	6.1 ± 0.6 ^bcd^	5.6 ± 0.4 ^cd^	4.9 ± 0.3 ^de^	<1 ^g^	<1 ^g^
(Y&M)	3.9 ± 0.3 ^e^	2	4.7 ± 0.5 ^de^	6.8 ± 0.3 ^abc^	6.7 ± 1.4 ^abc^	<1 ^g^	<1 ^g^
(log CFU/g)		3	5.6 ± 0.2 ^cd^	7.8 ± 0.3 ^a^	7.2 ± 0.5 ^ab^	<1 ^g^	<1 ^g^
		4	4.8 ± 0.3 ^de^	7.2 ± 0.3 ^ab^	7.0 ± 0.5 ^ab^	1.8 ± 0.9 ^f^	2.1 ± 2.0 ^f^

* Detection limit: 1-log CFU/mL. Values followed by the same letter (a–h) are not statistically different at *p* < 0.05.

**Table 2 foods-14-00540-t002:** Moisture content (%), pH, titratable acidity (TA, %), and total soluble solids (TSSs, %) of fresh and preserved strawberry. RF: refrigeration, RF + S: refrigeration with sucrose solution, RF + C: refrigeration with sucrose + CaCl_2_ + ascorbic acid (AA) solution, ICS + S: isochoric cold storage with sucrose solution, ICS + C: isochoric cold storage with sucrose + CaCl_2_ + AA solution.

Parameters	Fresh	Shelf Life (Weeks)	RF	RF + S	RF + C	ICS + S	ICS + C
		1	85.8 ± 2.6 ^abcd^	91.5 ± 0.7 ^a^	89.7 ± 1.1 ^a^	90.2 ± 2.7 ^a^	88.6 ± 0.8 ^ab^
Moisture (%)	89.7 ± 0.7 ^a^	2	85.0 ± 1.1 ^abcde^	86.1 ± 0.9 ^abc^	85.0 ± 0.3 ^abcde^	86.1 ± 0.6 ^abc^	85.8 ± 0.7 ^abcd^
		3	67.6 ± 4.6 ^f^	78.0 ± 1.6 ^e^	79.8 ± 2.7 ^cde^	79.3 ± 6.3 ^cde^	82.1 ± 0.4 ^bcde^
		4	58.7 ± 6.4 ^g^	61.6 ± 9.2 ^fg^	63.1 ± 5.5 ^fg^	79.0 ± 2.1 ^de^	68.1 ± 1.5 ^f^
		1	3.2 ± 0.0 ^cdef^	3.5 ± 0.0 ^ab^	3.2 ± 0.0 ^def^	3.3 ± 0.0 ^cde^	3.0 ± 0.1 ^gh^
pH	3.4 ± 0.0 ^abc^	2	3.3 ± 0.0 ^bcd^	3.5 ± 0.1 ^a^	3.1 ± 0.1 ^ef^	3.4 ± 0.1 ^abcd^	2.9 ± 0.0 ^ghi^
		3	3.3 ± 0.1 ^bcd^	3.4 ± 0.0 ^abc^	2.8 ± 0.1 ^ij^	3.0 ± 0.1 ^fg^	2.8 ± 0.0 ^hi^
		4	3.2 ± 0.1 ^def^	-	-	3.2 ± 0.2 ^cdef^	2.6 ± 0.3 ^j^
TA (%)	1.8 ± 0.2 ^ghi^	1	2.1 ± 0.4 ^efg^	1.3 ± 0.1 ^i^	1.4 ± 0.1 ^hi^	1.5 ± 0.1 ^hi^	1.5 ± 0.4 ^hi^
		2	3.1 ± 0.2 ^cd^	2.0 ± 0.1 ^fgh^	2.2 ± 0.2 ^efg^	2.1 ± 0.1 ^fg^	2.2 ± 0.1 ^efg^
		3	4.2 ± 0.4 ^a^	3.6 ± 0.3 ^bc^	4.0 ± 0.7 ^ab^	2.6 ± 0.1 ^ef^	2.7 ± 0.3 ^de^
		4	4.5 ± 0.0 ^a^	-	-	3.5 ± 0.2 ^bc^	3.3 ± 0.8 ^c^
TSS (%)	8.8 ± 0.3 ^gh^	1	9.6 ± 0.6 ^gh^	7.5 ± 0.1 ^h^	8.6 ± 0.3 ^gh^	8.3 ± 0.3 ^h^	8.9 ± 0.1 ^gh^
		2	12.9 ± 2.7 ^ef^	11.1 ± 0.6 ^fg^	13.6 ± 0.3 ^def^	12.5 ± 1.1 ^ef^	11.0 ± 1.6 ^fg^
		3	17.1 ± 1.1 ^c^	15.9 ± 2.0 ^cd^	15.6 ± 0.9 ^cd^	14.7 ± 2.7 ^cd^	14.5 ± 1.0 ^de^
		4	23.5 ± 2.0 ^b^	-	-	21.0 ± 1.9 ^b^	26.9 ± 0.4 ^a^

Values followed by the same letter (a–j) are not statistically different at *p* < 0.05.

**Table 3 foods-14-00540-t003:** Lightness (*L**), redness (*a**), yellowness (*b**), chroma (*C**), and hue angle (*h°*) of fresh and preserved strawberry. RF: refrigeration, RF + S: refrigeration with sucrose solution, RF + C: refrigeration with sucrose + CaCl_2_ + ascorbic acid (AA) solution, ICS + S: isochoric cold storage with sucrose solution, ICS + C: isochoric cold storage with sucrose + CaCl_2_ + AA solution.

Color Attribute	Fresh	Shelf Life (Weeks)	RF	RF + S	RF + C	ICS + S	ICS + C
		1	28.6 ± 3.4 ^b^	25.4 ± 2.2 ^b^	27.0 ± 2.1 ^b^	26.0 ± 1.3 ^b^	26.3 ± 1.3 ^b^
*L**	28.5 ± 2.0 ^b^	2	28.3 ± 2.4 ^b^	28.6 ± 2.4 ^b^	31.0 ± 2.2 ^b^	27.0 ± 1.9 ^b^	25.8 ± 2.9 ^b^
		3	29.4 ± 7.3 ^b^	32.8 ± 8.3 ^b^	25.5 ± 6.4 ^b^	25.7 ± 1.1 ^b^	27.0 ± 2.1 ^b^
		4	32.0 ± 9.0 ^b^	55.5 ± 13.7 ^a^	61.8 ± 13.9 ^a^	27.5 ± 1.3 ^b^	26.7 ± 2.7 ^b^
		1	14.7 ± 3.0 ^abc^	10.4 ± 2.9 ^cde^	14.6 ± 3.4 ^abc^	11.0 ± 1.0 ^cde^	11.5 ± 0.3 ^cde^
*a**	15.2 ± 3.1 ^abc^	2	14.0 ± 3.1 ^abc^	14.7 ± 3.2 ^abc^	19.4 ± 2.8 ^ab^	11.5 ± 1.9 ^cde^	15.5 ± 3.2 ^abc^
		3	11.9 ± 6.2 ^cd^	13.3 ± 6.8 ^abc^	19.5 ± 2.9 ^a^	11.8 ± 1.7 ^cde^	14.6 ± 3.4 ^abc^
		4	5.9 ± 5.4 ^def^	1.3 ± 0.9 ^f^	5.0 ± 4.7 ^ef^	12.6 ± 2.6 ^bcd^	19.2 ± 5.3 ^ab^
*b**	7.8 ± 2.1 ^abc^	1	6.8 ± 2.8 ^abcd^	3.8 ± 1.8 ^d^	5.8 ± 1.5 ^abcd^	3.7 ± 0.9 ^d^	5.0 ± 1.0 ^bcd^
		2	10.4 ± 3.6 ^a^	6.8 ± 1.2 ^abcd^	8.8 ± 1.3 ^abc^	4.9 ± 1.1 ^bcd^	6.1 ± 1.8 ^abcd^
		3	10.5 ± 2.9 ^a^	7.7 ± 2.6 ^abcd^	9.8 ± 3.5 ^ab^	4.7 ± 1.2 ^cd^	5.8 ± 1.5 ^abcd^
		4	8.8 ± 1.9 ^abc^	8.9 ± 1.6 ^abc^	10.0 ± 3.3 ^a^	6.0 ± 2.0 ^abcd^	9.3 ± 4.2 ^abc^
Chroma (*C**)	17.2 ± 3.2 ^abc^	1	16.3 ± 3.6 ^abcd^	11.2 ± 2.9 ^cd^	15.7 ± 3.6 ^abcd^	11.6 ± 1.0 ^cd^	12.6 ± 0.5 ^cd^
		2	17.7 ± 3.9 ^abc^	16.3 ± 3.3 ^abcd^	21.3 ± 2.9 ^ab^	12.5 ± 2.0 ^cd^	16.7 ± 3.6 ^abc^
		3	16.4 ± 5.2 ^abc^	15.6 ± 6.9 ^abcd^	21.9 ± 3.9 ^a^	12.7 ± 1.9 ^cd^	15.7 ± 3.6 ^abcd^
		4	11.2 ± 4.2 ^cd^	9.1 ± 1.5 ^d^	11.8 ± 4.1 ^cd^	14.0 ± 3.2 ^bcd^	21.3 ± 6.7 ^ab^
Hue (*h°*)	27.2 ± 6.8 ^def^	1	24.4 ± 6.8 ^def^	20.3 ± 9.5 ^ef^	21.6 ± 2.3 ^ef^	18.6 ± 4.1 ^f^	23.3 ± 4.4 ^def^
		2	36.1 ± 9.1 ^cd^	24.4 ± 4.8 ^def^	24.7 ± 2.8 ^def^	23.2 ± 4.3 ^def^	21.2 ± 2.5 ^ef^
		3	40.9 ± 12.5 ^c^	33.2 ± 12.6 ^cde^	26.0 ± 4.6 ^def^	21.4 ± 3.5 ^ef^	21.6 ± 2.3 ^ef^
		4	55.8 ± 16.6 ^b^	79.8 ± 6.8 ^a^	60.5 ± 17.9 ^b^	25.3 ± 3.0 ^def^	25.2 ± 2.7 ^def^

Values followed by the same letter (a–f) are not statistically different at *p* < 0.05.

**Table 4 foods-14-00540-t004:** Hardness (g) of fresh and preserved strawberry. RF: refrigeration, RF + S: refrigeration with sucrose solution, RF + C: refrigeration with sucrose + CaCl_2_ + ascorbic acid (AA) solution, ICS + S: isochoric cold storage with sucrose solution, ICS + C: isochoric cold storage with sucrose + CaCl_2_ + AA solution.

Hardness	Fresh	Shelf Life (Weeks)	RF	RF + S	RF + C	ICS + S	ICS + C
		1	237 ± 101 ^bcd^	123 ± 35 ^d^	185 ± 48 ^cd^	163 ± 30 ^d^	243 ± 81 ^bcd^
	194 ± 95 ^bcd^	2	142 ± 76 ^d^	150 ± 44 ^d^	177 ± 47 ^d^	160 ± 59 ^d^	218 ± 99 ^bcd^
		3	295 ± 126 ^bcd^	202 ± 133 ^bcd^	209 ± 111 ^bcd^	267 ± 42 ^bcd^	185 ± 48 ^cd^
		4	703 ± 589 ^ab^	273 ± 142 ^bcd^	416 ± 179 ^abc^	232 ± 41 ^bcd^	567 ± 194 ^a^

Values followed by the same letter (a–d) are not statistically different at *p* < 0.05.

## Data Availability

The original contributions presented in the study are included in the article, further inquiries can be directed to the corresponding author.

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
