# Peer review of "Application of Isochoric Impregnation: Effects on Microbial and Physicochemical Parameters and Shelf Life of Strawberries Stored Under Refrigeration"

_foods, 2025, doi:10.3390/foods14030540_

Round 1
Reviewer 1 Report
Comments and Suggestions for Authors The manuscript by Atci et al. investigated the application of isochoric impregnation on the shelf-life of cold-stored strawberry fruit, microbial, physicochemical, and quality attributes were studied. In my opinion, the manuscript can be accepted after minor revision. Please see my comments to improve the manuscript. -The authors and their group have extensive research on isochoric impregnation technology on a wide range of different fresh produce. My concern is for different fruit, regardless of different types, maturities, and varieties, it's difficult to utilize the fruit's initial quality. For fruit supercooling storage, that means different fruit may have different frozen temperatures. How did the authors ensure that the concentration of the solution and pressures used were suitable for the freezing point of the fruit to prevent freezing? -Another concern is the author claimed the control (RF group) was stored at 4 °C for 4 weeks. This means the fruit was stored in different conditions, the control group was kept in an ambient environment, whereas other groups were immersed in solutions for a certain period. Is the weight loss comparison reasonable? How much water/solute was absorbed by the fruit? -Line 93-95, please provide the criteria of the fruit. Such as the maturity of the fruit, the size, and the initial fruit quality. The authors also need to provide the method for cleaning and sterilization method. -Line 97-98, how many fruit were used for each treatment? How was the volume of the fruit. What kind of pouch? -Line 98-100, please provide the citation of selecting these solutions. -Were all the percentages in section 2.1 are all w/v? -Line 132-133, what was the relative humidity of the RF group? -Line 147, 2 to 3 grams of fruit is too little, which cannot represent the result, by different fruit sizes, shapes, and different surface areas. How much fruit were used in total for all the replicates? -Line 151, how much is "normal" speed? Is 60 seconds enough to wash all the microbes from the surface? -Line 160-164, the author did not classify the degree of fruit decay, only defined it as rotten and nonrotten. The author needs to propose improvement methods in subsequent discussions. Meanwhile, the author did not specify the number of fruits used in the overall experiment, and should rotten fruits be discarded? According to Figure 1, the decay rate of the RF group has reached 100% during the 4-week shelf life. So, is the microbial count in Table 1 still accurate? -Line 242-244, the authors did not describe the replications and repeat information of the experiment.Author Response
Please see the attachment.

Reviewer 2 Report
Comments and Suggestions for Authors
This work aims to use isochoric impregnation to extend the shelf life and quality propeties of strawberry. The experiments were sufficient and the results were well analyzed and presented. The obtained data could provide novel insghts in the field of produce preservation. Three minor points were provided for its improvement.
1. The standarlization should be followed through the whole MS, e.g., CaCl2 and E. coli.
2. The changes of microbial counts should be deeply discussed, in the point of view of mcirobial-microbial interaction and microbial ecology.
3. Stress and elicitor was responsible to induce the content of active compounds. The changes of anthocyanin and ascorbic acid should be discussed in a perspective of plant physiology.
Reviewer 3 Report
Comments and Suggestions for Authors
The work, which is very well structured and substantiated, is innovative, with good and up-to-date bibliographical research.
The methodology of isochoric impregnation during isochoric cold storage to extend the shelf life of fruit is not yet a widely used practice and involves innovative and advanced techniques that have been studied in recent years in the context of food preservation and quality control. This is evidenced by the bibliographic references cited, which are very recent, including some from 2024. This innovative method has the potential to improve food preservation without damaging its structure.
A good and clear presentation of the results, followed by a discussion that is always very well founded on studies carried out in this area for other fruits.
I would just like to point out that in this type of test the fruit should be processed immediately after harvest. As they were bought in a supermarket, we don't know exactly how long it took between harvesting and the start of the trials, nor do we know the handling conditions that need to be taken into account, as described in line 96. The duration of the trial is 4 weeks, and although all the fruit was in the same conditions for all the trials (T0 the same for all), this doesn't correspond to the actual time between harvest and the end of the trial. Perhaps it should be pointed out that it is not possible to start using the fruit immediately after harvesting, for example because it is impossible to buy it directly from the producer.
In point 2.3 Experimental protocol (lines 123-131), although it is very clear, it might be good to add a diagram to better visualise the test.
The bibliographical references cited throughout the text should be standardised, e.g. line 322 Bilbao-Sainz, Chiou, Takeoka, Williams, Wood, Powell-Palm, Rubinsky, & 322 McHugh, 2022 and line 319 Petriccione et al., 2015.
In the bibliographic references
Line 558 - Reference 12. Bilbao-Sainz, C., Millé, A., Chiou, B.-S., Takeoka, G., Rubinsky, B., & McHugh, T. (2024). Calcium impregnation during isochoric....
Line 567 Reference 15. Bilbao-Sainz, C., Olsen, C., Chiou, B. S., Rubinsky, B., Wu, V. C., & McHugh, T. (2024) Benefits of isochoric freezing for carrot ........
The two cited in the text as Bilbao-Sainz et al (2024) are not known to correspond, so one should be 2024a and the other 2024b.
Comments on the Quality of English LanguageVery clear and objective language
Reviewer 4 Report
Comments and Suggestions for Authors
This manuscript addresses an innovative and sustainable approach to extending the shelf life of strawberries through isochoric impregnation and cold storage. The research is well-framed with a clear objective and detailed methodology. I thoroughly enjoyed reviewing the manuscript. However, several aspects require improvement for better clarity, scientific rigor and overall presentation. A minor revision is needed before accepting the manuscript. Below are detailed comments with line numbers (of the submitted manuscript).
L(ine)28- How would you specify the measurement of quality as mentioned- “...maintained the quality of the strawberries and can effectively extend the shelf life of strawberries for up to 4 weeks.”? Clarify/add the parameters.
L40-45- Provide a citation for the claim regarding the shelf life of strawberries at room temperature.
L57-63: Add quantitative data or examples of cost comparisons between MAP and isochoric methods to support the claim.
L70-76: Include a brief explanation of the advantage of isochoric systems over conventional freezing because this will focus/ emphasize the the reduction in energy consumption.
L98-103- I suggest briefly explain the sensory testing methodology used to determine the acceptable concentration of CaCl2. This will add transparency to the experimental design.
L114-16- Please add details on how sterility was ensured for the water used in the isochoric chambers.
L272-279: Not clear what is meant by “...effectively kept microbial populations below the detection limit in strawberries for both solutions...”. Quantify the detection limit and include comparative microbial data from other studies.
L318- Relate this observation to the mechanisms of isochoric impregnation and its effectiveness in mitigating water loss. THis is very important and will strengthen the link between findings and the research objective.
L388- please add references to support the role of calcium in preserving color stability, and clarify whether this observation was statistically significant.
L466- Interesting observation- “...ICS samples showed significantly lower anthocyanin contents after one week. However, they exhibited significantly higher total anthocyanin contents at week 4...” Actually, I was looking for a scientific explanation of this outcome. Suggest discussing why anthocyanin levels initially decreased and later increased, considering storage conditions and molecular interactions.
L508- Wait! I am not sure what is meant by “...ICS effectively minimized weight loss during the refrigerated storage.” You need to reinforce the practical benefits of ICS with data. Did you quantify the weight loss reduction achieved through ICS and compare it with conventional methods?
L511- Revise- “Overall, ICS with sucrose, CaCl2, and ascorbic acid impregnation was the most effective...” You need to inspire further research and application. I suggest ending the conclusion with a forward-looking statement emphasizing the potential for commercialization or scaling up this technology.
